# Cyberbullying Perpetration and Socio-Behavioral Correlates in Italian and Spanish Preadolescents: A Cross-National Study and Serial Mediation Analysis

**DOI:** 10.3390/ijerph22030389

**Published:** 2025-03-07

**Authors:** Gianluca Mariano Colella, Rocco Carmine Servidio, Anna Lisa Palermiti, Maria Giuseppina Bartolo, Paula García-Carrera, Rosario Ortega-Ruiz, Eva M. Romera

**Affiliations:** 1Department of Cultures, Education and Society, University of Calabria, Via Pietro Bucci, Arcavacata Di Rende, 87036 Cosenza, Italy; servidio@unical.it (R.C.S.); annalisa.palermiti@unical.it (A.L.P.); mariagiuseppina.bartolo@unical.it (M.G.B.); 2Department of Psychology, Universidad de Córdoba, San Alberto Magno s/n, 14071 Córdoba, Spain; m62gacap@uco.es (P.G.-C.); ortegaruiz@uco.es (R.O.-R.); eva.romera@uco.es (E.M.R.)

**Keywords:** cybervictimization, cyberbullying perpetration, general aggression model, moral disengagement, preadolescence, problematic social media use

## Abstract

The spread of information and communication technologies (ICTs) has brought advantages and disadvantages, particularly impacting youth, who use the Internet and social media applications daily. In preadolescents’ social development, problematic social media use (PSMU) and cyberbullying (CB) are potential risk factors across several countries. PSMU is defined as the lack of regulation of social media platforms that is associated with negative outcomes in everyday life, while CB refers to using digital technology to harass, threaten, or embarrass another person. Among preadolescents, CB perpetration is frequently associated with cybervictimization (CV) experiences. The underlying mechanisms that drive this relationship have received limited attention. The aim of the cross-national comparative study, rooted in the general aggression model, is to investigate the direct and indirect effects between cyberbullying perpetration and cybervictimization, testing a model involving PSMU and moral disengagement (MD) as serial mediators in this association. A total of 895 Italian and Spanish preadolescents (M_age_ = 11.23, SD_age_ = 1.064) completed a self-report survey during school hours. Descriptive statistics were computed, and a serial mediation model was run. The results show that CV is positively associated with CB, and that PSMU and MD positively serially mediate the CV–CB link. This study’s insights suggest the need for tailored educational interventions targeting European youth, to promote more positive online social interactions and a safer digital environment.

## 1. Introduction

### 1.1. Cybervictimization as a Predictor of Cyberbullying Perpetration

In recent decades, Western countries have faced the spread of information and communication technologies (ICTs) and social media applications [1], which has increased the number of individuals connected daily to the Internet for several activities [2]. Through digital devices, preadolescents engage in positive and negative behaviors and habits, encountering opportunities and risks previously beyond reach [3].

Among the risks associated with problematic Internet use and problematic social media use (PSMU) in preadolescence is cyberbullying, a social problem progressively studied as a digital evolution of traditional bullying [4]. Cyberbullying perpetration is defined «as the use of digital technologies to harass, intimidate, or embarrass others» [5,6]. The authors reviewed multiple studies and determined that, on a global scale, the average annual rate of cybervictimization ranges from 14% to 21% [7]. Between 10% and 72% of youths have reported experiencing cyberbullying [8]. These alarming rates are associated with severe consequences for victims, including mental health issues, social development difficulties, and diminished self-esteem [9,10]. Additionally, cybervictimization refers to being harmed or mistreated via electronic communication technologies such as the Internet and social media platforms, encompassing aggression, harassment, or intimidation perpetrated online [11], and is expressed in multiple forms, including harassment, impersonation, stalking, or spreading malicious rumors or images [12]. It may cause emotional distress, psychological harm, and general long-term negative consequences for victims’ personality development [13].

In the European context, the prevalence of cyberbullying perpetration and cybervictimization among young individuals has garnered increasing attention, shedding light on the scope of this issue within this diverse setting in which multiple cultures participate. Italy and Spain, as evidenced by the proliferation of studies on the topic [14], are among the European countries most affected by these social mechanisms. In the 2022 Italian Health Behavior in School-aged Children (HBSC) survey, 94,178 children aged 11, 13, 15, and 17 completed the questionnaire, which was distributed throughout all regions; 6388 classes were sampled. In the 11-year-old age group, 17.2% of males and 21.1% of females have been identified as victims of cyberbullying; 12.9% of boys and 18.4% of girls were victims of cyberbullying perpetration; 9.2% of boys and 11.4% of girls were reported to be victims in the 15-year-old age group. The survey reports that in families, as age increases, the ease with which boys open up to their parents decreases; 13- and 15-year-old girls, compared to boys of the same age, find it more difficult to talk to their father figure. In 2022, 31% of minors were cybervictims at least once, compared to 23% in 2020. In Spain, a systematic review of the literature, including 21 studies, revealed that, since 2010, the median perpetration of cyberbullying in Spain was 24.64%, with a median cybervictimization prevalence of 26.65% [15,16].

Developmental research dealt with the link between cyberbullying perpetration and cybervictimization, revealing their bidirectional association [17]. Globally, most cybervictimized preadolescents subsequently suffer from mental health issues, including increased stress and the adoption of maladaptive coping strategies [18]. Further research indicates that individuals who experience cybervictimization are more likely to engage in cyberbullying perpetration themselves [19]. This phenomenon can be understood through the lens of the “cycle of violence” theory, which suggests that victims of aggression may resort to aggressive behaviors as a coping mechanism or to regain a sense of control and power [20], suggesting that cybervictimization experiences can be viewed as situational factors triggering negative emotional responses, such as anger or distress [21]. Relatively few studies have examined those who are both victims of cyberbullying and perpetrators (i.e., cyberbully-victims) [22]. In the digital context, the anonymity and perceived detachment provided by online interactions can exacerbate this cycle, making it easier for victims to become perpetrators without social repercussions, in association with emotional distress, social isolation, and the desire for revenge [23,24]. Understanding this link is crucial for developing effective prevention and intervention strategies targeting both the victims’ needs and the motivations behind perpetration, aiming to break this detrimental cycle and promote healthier online interactions among preadolescents [25].

Among the possible risks faced by some preadolescents, PSMU, video games, and Internet use have progressively emerged as potentially pressing public health concerns across numerous Western countries due to their impact on preadolescents’ social development processes [26,27,28,29,30,31]. In particular, in association with parental monitoring attitudes, recent studies have illustrated the interplay between escapism as an attempt to reduce negative emotions, parental mediation styles, and interpersonal skills [32,33,34]. Building upon prior research [35], the current study posits that cybervictimization in preadolescence might serve as a precursor to cyberbullying perpetration, reflecting the cyclical nature of online aggression [36]. In the context of cyberbullying experiences, victims are often driven by a personal need to regain control or seek revenge on the bully and retaliate by becoming perpetrators themselves [37]. Therefore, cybervictimization experiences might significantly increase the likelihood of individuals engaging in subsequent cyberbullying perpetration, perpetuating a vicious cycle of online aggression and intensifying the psychological distress experienced by both victims and perpetrators [38]. This study aims at uncovering the underlying processes involved in the cybervictimization–cyberbullying perpetration link in preadolescence, introducing the serial mediating action played by PSMU and moral disengagement.

### 1.2. Theoretical Framework

The general aggression model (GAM) describes how personal and situational factors could interact to affect an individual’s thoughts, feelings, and behaviors, leading to aggressive conduct [39]. According to the model, aggressive behaviors are the result of complex interactions between situational variables (e.g., provocation and exposure to violent media) and individual differences (e.g., personality traits and past experiences), which influence cognitive (e.g., aggressive thoughts), affective (e.g., anger), and arousal states. These internal states, in turn, might affect decision-making processes, which can result in aggressive or non-aggressive behaviors depending on the context and the individual’s control over their responses [40].

The GAM offers a comprehensive framework for understanding cyberbullying among preadolescents, integrating individual-specific and situational factors [41]. Specifically, the model suggests that experiences of cyberbullying alter an individual’s state, prompting aggressive behaviors. Cybervictimization could serve as a situational trigger, catalyzing cyberbullying, and aversive events could activate hostile attitudes, leading to aggressive impulses as situational triggers [42].

While previous studies have examined the relationship between traditional bullying and cyberbullying and the influence of factors such as gender, emotional problems, depression, and anxiety [43], the underlying mediating mechanisms between cybervictimization and cyberbullying perpetration still remain unclear. The GAM, addressing both person-specific and situation-specific factors, provides valuable lens through which to explore these mediating mechanisms. Within the GAM, PSMU and MD might be interpreted as serial mediating factors in the cyberbullying perpetration–cybervictimization link, providing a deeper understanding of the mechanisms through which cybervictimization leads to the perpetration of cyberbullying in preadolescence. The proposed pathway suggests that experiencing cybervictimization may elevate PSMU, leading to maladaptive user interactions with social media platforms. This study introduces PSMU as a mediator between CV and CB, a behavioral pattern that typically develops over a longer period. This, in turn, can encourage MD processes, by which people distance themselves from feelings of guilt or moral responsibility. These cognitive self-regulatory strategies eventually pave the way for engaging in cyberbullying behaviors, reinforcing the cycle of online aggression [44].

### 1.3. The Mediating Role of Problematic Social Media Use

Preadolescents are a vulnerable age group aged between 9 and 14 [45], in which the brain is still developing, and social relationships are essential for positive development [46]. As contemporary preadolescents grow up using digital tools and social media platforms from an early age, most of them today face both opportunities, such as for socialization, learning, and entertainment [47], and risks, such as PSMU, exposure to inappropriate content, and problematic online gaming, among others [48,49]. PSMU refers to the excessive or maladaptive individual patterns of engagement with social media platforms, characterized by withdrawal symptoms, mood modification, and conflict [50]. Thus, technological environments and social media platforms can foster social connections and provide platforms for self-expression and creativity [51]. They enable young people to access vast amounts of information, collaborate on educational projects, and build supportive online communities [52].

In this study, we adopted the term PSMU over “social media addiction” to avoid controversies, since the latter is not officially recognized as a disorder in the DSM-5 [53]. This decision aligns with the current literature, which distinguishes between problematic usage patterns and clinically diagnosable addiction [54]. For some individuals, this problematic habit can become a dominant activity in their lives, causing preoccupation (salience) and the usage of social media to alter moods or induce pleasurable feelings (mood modification). Over time, more engagement is needed to achieve the same effects (tolerance), and discontinuing use can result in negative psychological and sometimes physiological symptoms (withdrawal), often leading to relapse. This problematic use can cause internal conflicts, such as the loss of control, and external conflicts, including relationship and academic or work problems [55]. In the context of cyberbullying research, PSMU might play a mediating role in the cybervictimization–cyberbullying perpetration link for several reasons. Preadolescents who are frequently online may be more likely to encounter aggressive behaviors or become targets of harassment [56]. Moreover, PSMU can impair emotional regulation and coping mechanisms, making it more difficult for victims to manage the distress caused by cybervictimization experiences [57]. This emotional turmoil can lead some victims to engage in retaliatory aggression as a maladaptive coping strategy, thereby perpetuating the cycle of cyberbullying [58]. Additionally, PSMU involves seeking validation and social approval, which can exacerbate feelings of rejection and isolation when victimized, leading to aggressive behaviors to regain control and social standing [59]. Thus, the nature of PSMU can both increase the likelihood of experiencing cybervictimization and heighten the propensity to respond with cyberbullying behaviors [60]. In the context of the GAM, PSMU might be viewed as a personal maladaptive situational experience triggered by both environmental and individual variables, further leading to negative online social behaviors, such as cyberbullying perpetration. Thus, the role of PSMU in the CV–MD–CB link reflects its capacity to drive the progression of harmful online behaviors and amplify the impact of underlying moral attitudes.

### 1.4. The Mediating Role of Moral Disengagement

Moral disengagement (MD) entails cognitive self-regulatory mechanisms allowing individuals to rationalize or justify harmful behaviors, thereby alleviating feelings of guilt or moral responsibility, through cognitive restructuring, diffusion of responsibility, or euphemistic labeling [61]. These strategies can occur in multiple contexts, including cyberbullying episodes among preadolescents, where perpetrators may practice rationalizations, such as advantageous comparison, minimizing the harm caused, or using social comparison to justify their behaviors [62]. In the context of cyberbullying perpetration, moral disengagement is defined as “an influence on traditional bullying and cyberbullying cognitive process, by which a person justifies his/her harmful or aggressive behavior, by loosening his/her inner self-regulatory mechanisms […] which usually keeps behavior, in line with personal standards” [63] (p. 81). Regarding online social behaviors in preadolescence, MD mechanisms can contribute to the perpetration of a wide range of antisocial actions [64]. Cyberspace grants anonymity and reduced accountability, making it easier for individuals to engage in cyberbullying and aggressive online actions without facing any consequences [65], or to retaliate against perpetrators of online aggression. MD enables individuals to justify their actions and disregard the ethical implications of behaviors: “the advent of the Internet ushered in a ubiquitous vehicle for disengaging moral self-sanctions from transgressive conduct. The Internet was designed as a highly decentralized system that defies regulation. Anybody can get into the act, and nobody is in charge” [66] (p. 68).

Building on previous research, in the current study, MD is proposed as a mediating factor between cybervictimization, PSMU, and cyberbullying perpetration [67]. It may act as a link between PSMU and cyberbullying perpetration, suggesting that heightened PSMU might lead to the adoption of MD mechanisms, which in turn might more likely be associated with cyberbullying perpetration. MD involves cognitive restructuring and justifications to disengage from moral standards, allowing individuals to rationalize their harmful actions. These strategies might lead to perpetuating negative behaviors toward others in cyberspace [68]. Preadolescents might experience reduced self-awareness and restraints, fueling MD and subsequently influencing cyberbullying and cybervictimization dynamics. Within the GAM, MD processes might play a role in the individual desensitization that facilitates the perpetration of harmful online actions in cybervictims and problematic young social media users.

### 1.5. The Current Study

The current study investigates online social behaviors and psychological characteristics in Italian and Spanish preadolescents, linking cybervictimization, PSMU, MD, and cyberbullying perpetration. Moreover, the serial mediating model aimed at testing the effects of PSMU and MD on the association between cybervictimization and cyberbullying perpetration has been examined. The relationship between PSMU and MD is based on the premise that a generalized problematic engagement with social media might precede and facilitate disengaged moral reasoning, rather than the other way around. Specifically, PSMU may act as a precursor to MD, as unregulated social media use can foster a detachment from consequences, the normalization of harmful behaviors, and reduced self-regulation in online social interactions. In addition, the MD–PSMU causal relationship has been less explored in psychological research and was not an aim of the current study. We were interested in exploring the following research questions:Research Question 1: Is cybervictimization directly and positively associated with cyberbullying perpetration among Italian and Spanish preadolescents?Research Question 2: Is this relationship mediated by the serial indirect and positive effect of PSMU and MD in both samples?

As serial positive mediating effects, PSMU and MD may contribute to cyberbullying perpetration by facilitating negative and aggressive online interactions among peers, providing a platform for anonymity and disinhibition, and amplifying peer influence dynamics (see Figure 1). Thus, understanding the serial mediating roles of PSMU and MD is crucial for designing effective preventive interventions aimed at mitigating the negative impact of cyberbullying and cybervictimization among preadolescents.

Among the risk factors associated with cybervictimization and cyberbullying perpetration in preadolescence, it is worth investigating the interplay of PSMU and MD within the context of negative social online interactions among peers due to their pervasive influence in preadolescents’ social behaviors [69,70]. Understanding the dynamics between these variables in the link between cybervictimization and cyberbullying perpetration is essential for unraveling the processes that might enhance the cycle of negative online interactions among preadolescents [71]. Previous research conducted in Italy and Spain [72,73] highlights both sociocultural similarities and differences between the two countries regarding cyberbullying and cybervictimization. Given these parallels and distinctions, as well as comparable educational contexts [74], further investigation into the perpetuation of online aggression is crucial.

The hypotheses under investigation in this cross-national study are the following: (i) cybervictimization and cyberbullying perpetration present a positive direct association in a cross-national sample of Italian and Spanish preadolescents; and (ii) PSMU and MD positively and serially mediate the relationship between cybervictimization and cyberbullying perpetration in Italian and Spanish preadolescents.

These hypotheses are grounded in several factors. Firstly, the literature shows that PSMU has been linked to negative online behaviors among preadolescents, including cyberbullying, suggesting that individuals who engage in PSMU patterns may be more prone to experiencing cybervictimization or perpetrating cyberbullying [75,76]. Moreover, MD mechanisms, enabling individuals to justify or excuse antisocial actions, may play a crucial role in facilitating the transition from cybervictimization to cyberbullying perpetration by attenuating online moral inhibitions and increasing the likelihood of engaging in aggressive behaviors against peers [77,78]. The decision to employ a serial mediation model, wherein PSMU precedes MD in the direct relationship between cybervictimization and cyberbullying perpetration, stems from theoretical considerations and is supported by the existing literature [79]. PSMU might lead to negative engagement with social media platforms, potentially leading to the frequent adoption of dysfunctional social and cognitive attitudes and behaviors in daily life [80]. Previous research suggests that prolonged and passive use of social media applications may contribute to the development of negative attitudes and behaviors, including MD mechanisms [81], exacerbating the scope and negative consequences of the relationship between cybervictimization and cyberbullying. This approach tentatively addresses a gap in the literature concerning the interplay of PSMU and MD within the social dynamics of experiencing and perpetrating cyberbullying during preadolescence.

Lastly, the cross-national analysis involving Italy and Spain is motivated by the scientific urge to comprehend the psychosocial characteristics of online social behaviors among Italian and Spanish preadolescents [82], given that past studies highlighted similar trends regarding cyberbullying, as well as related factor levels in these two countries [83,84]. By delineating these relationships in the current cross-national study involving Italian and Spanish preadolescents, tailored interventions to target specific risk factors and promote healthier online behaviors within a European policy framework, fostering a safer and more positive digital environment for young individuals, can be proposed [85].

## 2. Materials and Methods

### 2.1. Participants and Procedures

Through a self-report survey, levels of cybervictimization, PSMU, MD, and cyberbullying perpetration were assessed in a convenience sample of Italian and Spanish preadolescents (*N* = 895, 54.6% Italian participants, 45.4% Spanish participants) aged 9–14 (*M* = 11.23, SD = 1.064) (see Table 1). All approached students participated in the survey in both countries since the excluded individuals did not receive consent from their parents.

This study and its procedures were conducted according to the criteria of the Ethics Committee of the University of Calabria (Prot. no. 0010986 of 13 February 2023). In Spain, this study and its procedures were preliminarily approved by the local research team and by the doctoral committee of the University of Cordoba. The procedures used in this study adhere to the tenets of the Declaration of Helsinki. In Italy, the participants were contacted after selecting thirty-one classes in schools in the region of Calabria (southern Italy) who were willing to participate. In both Italy and Spain, the schools were selected by using a search database in which a list of local school institutions was stored and persuaded to join the investigation through a motivation letter to the school principals introducing the purposes of this study. After obtaining permission from the corresponding school principals, the participants’ parents were informed by letter about the purpose of the research, the voluntary nature of participation, and the anonymity of responses. Parents provided informed consent for their son or daughter’s participation. In addition, participants provided signed assent agreeing to take part in this study. Italian and Spanish research assistants collected the data after providing a general description of the research aims and measures (defining social media and cyberbullying in terms of aggression and victimization). Participants had about 30 min to complete a self-report survey during class time and could withdraw at any moment.

### 2.2. Measures

The participants completed a self-report questionnaire containing a set of different measures. For the purposes of this study, only some of these measures were considered.

Sociodemographic information was collected, such as gender, age, and frequency of use and habits related to the Internet and social media platforms.

Problematic social media use was assessed using the Italian [86] and Spanish versions [87] of the Bergen Social Media Addiction Scale (BSMAS). The BSMAS is a 6-item self-report measure that assesses addictive behaviors related to social media use, such as withdrawal symptoms, mood modification, and conflict (e.g., “You spend a lot of time thinking about social media or planning how to use it”.). Participants rated each item on a Likert scale from 1 (*very rarely*) to 5 (very often), with higher scores indicating higher levels of PSMU (Italian sample: Cronbach’s α = 0.719; Spanish sample: Cronbach’s α = 0.760). The English version of the scale is provided in Appendix A [88].

Moral disengagement was measured using the Italian [89] and Spanish adolescent versions [90] of the *Moral Disengagement Scale* (MDS). The scale is a 24-item self-report measure that assesses cognitive mechanisms that allow individuals to justify or excuse their harmful behaviors (e.g., “It is alright to fight to protect your friends”.). Participants rated each item on a Likert scale from 1 (not at all) to 5 (very much), with higher scores indicating higher levels of moral disengagement (Italian sample: Cronbach’s α = 0.839; Spanish sample: Cronbach’s α = 0.806). Given confirmation from prior studies of the internal consistency of the single-factor structure of the scale [91], a total mean score was calculated for moral disengagement. The English version of the scale is provided in Appendix A [92].

Cyberbullying perpetration and cybervictimization were measured through the European Cyberbullying Intervention Project Questionnaire (ECIP-Q) in its Italian [93] and Spanish versions [94]. The ECIP-Q is a 22-item self-report measure, with the former 11 items measuring victimization and the latter 11 measuring aggressions, that assesses the frequency and severity of cybervictimization and cyberbullying behaviors in the past two or three months, such as spreading rumors, posting mean comments (e.g., “I altered pictures or videos of another person that had been posted online”.), or excluding others from online groups or stealing someone’s identity and personal information (e.g., “Someone created a fake account, pretending to be me”.). Participants rated each item on a Likert scale from 1 (never) to 5 (several times a week), with higher scores indicating higher levels of cybervictimization (Italian sample: Cronbach’s α = 0.846; Spanish sample: Cronbach’s α = 0.830) and cyberbullying (Italian sample: Cronbach’s α = 0.853; Spanish sample: Cronbach’s α = 0.616). The English version of the scale is provided in Appendix A [94].

### 2.3. Analysis Procedure

The statistical analyses were conducted in four steps using IBM SPSS 27.0. First, descriptive analyses (means and standard deviations) were computed alongside ***t***-tests for independent samples to examine gender differences. A van der Waerden’s data ranking transformation on the kurtosis and skewness values of cybervictimization and cyberbullying perpetration was applied due to their excessively high magnitudes. This transformation aimed to normalize the distribution and mitigate the extreme values observed in these variables. Effect sizes were measured adopting Cohen’s *d*, providing a clearer understanding of gender influences on the variables. Second, Pearson bivariate correlations were analyzed to explore potential relationships between the key variables. Third, a serial mediation analysis (using SPSS Process macro v.27.0, Model 6) [95] examined whether PSMU and MD serially mediated the association between CV and CB. This approach utilized bootstrapping with 5000 resamples, allowing for more accurate confidence intervals and reducing the risk of type I errors. The mediation analysis also assessed the total effect and proportion mediated (PM) to gauge the indirect effect’s contribution.

A significant effect is indicated when the confidence intervals do not encompass zero. Previous studies found that gender was associated with cyberbullying perpetration and cybervictimization [96,97,98]. Thus, this variable was included as a covariate in all analyses. Gender was dichotomous (0 = female, 1 = male, 2 = I prefer not to answer, 3 = other). Cases where participants selected 2 or 3 were excluded from the analysis due to their very low numbers, which did not impact the overall results. Additionally, age and nationality were also included as covariates to account for their potential influence on the variables’ outcomes.

## 3. Results

### 3.1. Descriptive Statistics

In Table 1, descriptive statistics of the variables are reported for both Italian and Spanish participants, showing overall low levels of cybervictimization and cyberbullying perpetration.

Table 2 summarizes the results of the independent samples *t*-test conducted on the investigated variables, categorized by gender.

The results indicate no statistically significant mean differences across the primary variables when comparing gender groups, with most effect sizes reflected by small to medium Cohen’s *d* values. This suggests that, in general, gender did not play a significant role in differentiating levels of PSMU, cybervictimization, or cyberbullying. However, the analysis revealed that MD was the only variable to show a medium effect size, hinting at some degree of gender-related variance. Specifically, males showed higher mean scores in both MD and cyberbullying (*p* < 0.05), suggesting that they may be more involved in aggressive online interactions than females. This finding may imply that males and females differ slightly in how they rationalize or justify aggressive behaviors online, warranting further investigation.

Table 3 summarizes the results from the bivariate correlations among the variables, indicating strong positive correlations between all investigated variables. Notably, PSMU presented strong correlations with MD, CV, and CB, and CV presented a strong positive correlation with CB. We included age in the correlation analysis because we were interested in exploring whether problematic social media use increases with age.

### 3.2. Mediation Model

Mediation analysis was performed to explore the serial mediating role of PSMU and MD in the association between cybervictimization and cyberbullying perpetration. The results (Table 4) reveal a significant total effect of CV on CB perpetration (β = 0.071, SE = 0.013, 95% CI [0.045; 0.098]), indicating that higher levels of CV are associated with increased CB perpetration.

Furthermore, PSMU and MD serially mediate this relationship, showing strong positive associations. Specifically, CV predicts PSMU (β = 0.35, 95% CI [1.68; 2.38]), which in turn predicts MD (β = 0.28, 95% CI [0.59; 0.93]). MD subsequently predicts CB perpetration (β = 0.14, 95% CI [0.004; 0.011]), suggesting that PSMU and MD play crucial roles in the pathway from CV to CB. The results highlight a significant indirect effect of PSMU and MD on the cybervictimization–cyberbullying perpetration link (H2: PSMU and MD positively and serially mediate the relationship between cybervictimization and cyberbullying perpetration in Italian and Spanish preadolescents).

In the mediation model, cybervictimization positively predicted PSMU, MD, and cyberbullying. Further, both PSMU and MD presented statistically significant indirect serial effects in the cybervictimization–cyberbullying association. The results show that PSMU and MD positively and partially mediated the relationship between cybervictimization and cyberbullying perpetration (see Figure 2).

## 4. Discussion

This study aimed (1) to examine the positive associations between CV, PSMU, MD, and CB perpetration in a sample of Italian and Spanish preadolescents and (2) to examine the positive serial mediating action of PSMU and MD on the CV–CB link.

In line with H1 (cybervictimization and cyberbullying perpetration present a positive direct association in a cross-national sample of Italian and Spanish preadolescents), the results confirm our hypothesis and reveal that the scores for the primary variables mirror findings from previous studies [73]. These findings reinforce the understanding of cyberbullying as a potential global concern for preadolescents, emphasizing the significant roles of cybervictimization and PSMU in both Italian and Spanish participants. From the perspective of the GAM, various situational and individual factors, such as PSMU and MD, intensify negative online behaviors among preadolescents [99]. These variables might exacerbate aggressive tendencies, making them central to understanding the complex interplay between victimization and perpetration in online social peer interactions [100]. By capturing these dynamics in both Italian and Spanish contexts, the current findings offer an interesting cross-national perspective on the cyclical nature of online aggression, highlighting how these factors shape harmful social media interactions among preadolescents. The increasing integration of social media applications’ use in preadolescents’ daily life worldwide through smartphone devices amplifies the likelihood of cybervictimization, especially among those heavily engaged with these platforms [101]. These findings highlight the need for targeted prevention and intervention strategies to mitigate these risks. Addressing such issues requires early identification and tailored support, particularly in regions with heightened exposure to digital media.

Cultural, educational, and policy-driven factors may influence the differences between Italian and Spanish participants in cyberbullying-related variables. One key difference lies in educational policies regarding technology use. Such policy variations between Italian and Spanish schools may contribute to differences in PSMU and online aggression patterns [102]. Beyond policy, cultural attitudes toward digital supervision and parenting styles may also play a role. Research suggests that Mediterranean parenting styles, while traditionally protective, vary across countries in terms of digital monitoring. Spanish parents tend to emphasize structured family routines and digital literacy, whereas Italian parents may lean toward greater autonomy in online interactions [103]. Future research should further explore these cultural moderators, considering how national policies, educational strategies, and familial norms interact to influence cyberbullying dynamics in different contexts, as well as motivations driving online behaviors [32,33,34].

The results from the mediation analysis provide empirical evidence supporting the serial mediation model, confirming H2 (PSMU and MD positively and serially mediate the relationship between cybervictimization and cyberbullying perpetration, such that higher levels of cybervictimization lead to higher levels of PSMU and MD, which in turn lead to higher levels of cyberbullying, both in Italy and Spain). PSMU and MD serially mediate the relationship between CV and CB in both Italy and Spain. The findings reveal that experiencing cybervictimization might promote problematic engagement with social media applications in Italian and Spanish participants, which in turn could increase MD tendencies, ultimately leading to higher levels of cyberbullying perpetration. This cyclical pathway underscores the role of PSMU in amplifying MD, illustrating that a broader negative engagement with social media platforms strengthens the risk of adopting morally disengaged behaviors that justify harmful online actions [104]. The model fitting within the GAM framework demonstrates that cybervictimization might act as a situational factor that might trigger self-regulatory cognitive and emotional changes in participants, making them more likely to engage in maladaptive social media use [105]. This increased use of social media platforms might provide fertile ground for developing MD mechanisms, where participants may justify or excuse their own aggressive actions. This, in turn, perpetuates the cycle of cyberbullying, especially for those already victimized online [106]. The stronger indirect effects through PSMU further confirm that frequent and problematic use of social media strengthens aggressive scripts and attitudes, leading to a higher likelihood of cyberbullying [107]. The cross-national aspect of this study, involving Italian and Spanish participants, highlights the importance of these findings across different contexts. Despite similarities between these countries, such as language and cultural values, differences in smartphone use policies at schools can affect the intensity and frequency of social media platform engagement and thus online behaviors. The serial mediation link identified in both contexts reinforces that national variations do not diminish the robust effect of PSMU and MD on cyberbullying perpetration.

According to the GAM, aggression is the result of a complex interaction between individual and situational variables [108]. In this study, PSMU and MD might represent situational inputs and person-specific factors that influence the internal state of Italian and Spanish preadolescents, triggering aggressive responses such as cyberbullying. The results from this study reinforce the GAM’s premise that while situational factors (e.g., cybervictimization) are critical in initiating aggression, the long-term perpetuation of these behaviors is deeply rooted in personal and cognitive factors (e.g., moral disengagement) amplified by external stimuli (such as problematic engagement with social media) [109]. These results point to the importance of addressing both environmental (e.g., regulation of social media use) and cognitive (e.g., moral reasoning) factors in intervention strategies, supporting the GAM’s multifaceted approach to understanding aggression.

## 5. Conclusions

The findings of this study contribute to the literature on cyberpsychology, extending knowledge of the risk factors connected to cyberbullying and its perpetration in European preadolescents. In addition, the results underscore the need for a common European policy framework promoting guidance on responsible and pro-social digital literacy citizenship to prevent cyberbullying and negative online social experiences in preadolescence. By providing research insights to inform tailored interventions, this study aims to promote a safer online environment and enhance the well-being of Italian and Spanish preadolescents.

Based on our findings and the previous literature, educators and policymakers might consider adopting evidence-based research interventions to practically address cyberbullying and its perpetration, as well as negative online habits in preadolescents and adolescents, through socio-emotional and moral learning, peer culture education, or digital literacy [106,107,108].

This research is cross-nationally significant, contributing to the understanding of cyberbullying within the Italian and Spanish contexts and guiding evidence-based strategies to address its impacts. The serial mediation model illuminates how PSMU and MD serially and partially mediate the pathway from cybervictimization to cyberbullying, offering a nuanced understanding of these dynamics. The cross-national nature of the sample strengthens this study’s relevance, indicating that despite slight cultural variations, such as smartphone policies in schools, the core relationships between these variables persist across contexts. This research underscores the importance of culturally tailored intervention strategies targeting online behaviors and cognitive distortions early in preadolescence. Its findings can inform the development of preventive measures and policies in schools, both within Europe and internationally, thereby expanding the scope of cross-cultural cyberbullying research.

In sum, this research emphasizes the critical role of PSMU and MD in the cycle of cyberbullying, particularly for those who have experienced cybervictimization, and highlights how interventions targeting social media application use and cognitive self-regulation could break the escalation of online aggression across different European contexts. This work moves the field of cyberpsychology forward by highlighting the need for comprehensive interventions that address behavioral and cognitive factors contributing to cyberaggression, stressing the critical period of preadolescence for prevention efforts. It also emphasizes the importance of cross-national research, showcasing how country-specific factors, such as school policies, shape online behaviors in distinct yet comparable ways. Through this, this study advocates for European policies to effectively address cyberbullying.

## 6. Limitations and Further Research

This study’s reliance on self-report data introduces possible biases such as social desirability and recall bias, which could affect the accuracy of participants’ responses. One significant limitation of the present study lies in its reliance on cross-sectional data to test a serial mediation model, which inherently assumes causal relationships between variables. While mediation analysis is a useful statistical technique, it does not allow robust testing of causality when bidirectional and simultaneous influences likely exist among the variables. Specifically, in the relationship between CV and CB, there is strong theoretical justification for conceptualizing cybervictimization as a situational trigger that elicits impulsive aggressive behaviors, a mechanism often observed in aversive situations where hostile attitudes are activated. This view suggests a more immediate, reactive link between CV and CB. The role of PSMU may create tension within the hypothesized temporal sequence, as impulsive aggressive behaviors theoretically contradict the sustained behavioral engagement associated with PSMU. However, the model and relationships tested, and the results obtained, confirm similar results obtained from longitudinal studies on these variables [35]. Despite these challenges, the present study sought to address this complex relationship in a novel way by exploring PSMU as a mediating mechanism in a cross-national sample of Italian and Spanish preadolescents. While causal claims cannot be made, this approach sheds light on potential underlying processes that link victimization and perpetration, emphasizing PSMU’s pervasive role in shaping digital interactions. However, future research should employ longitudinal data to better assess the directionality of these relationships. By comparing two European cultural contexts, this study offers insights into shared and unique dynamics among preadolescents in both countries, providing a foundation for future longitudinal research to better disentangle the temporal and bidirectional relationships between CV, PSMU, and CB [110,111,112].

Another significant limitation is the cross-national nature of the sample drawn from Italy and Spain. Despite their geographic proximity, these countries exhibit differences in school policies, notably the use of smartphones. In Spanish schools, smartphones were rarely used, which may influence the frequency and context of problematic social media use, compared to Italy, where more permissive policies exist. These differences could affect how students interact with social media, potentially shaping their experiences of cybervictimization, moral disengagement, and cyberbullying differently. Cultural factors such as norms around technology use and educational practices were not directly measured, limiting the understanding of how contextual factors influence the findings. Moreover, the lower reliability of the ECIP-Q among Spanish preadolescents on the cyberbullying perpetration items may be influenced by these cultural tendencies, affecting online aggressive behavior and general normative habits regarding smartphone and Internet use. Furthermore, the use of Likert scales to measure complex constructs such as moral disengagement might not fully capture the depth and variability of this phenomenon, raising concerns about the precision of measurements. Future research should explore longitudinal designs to understand the evolving nature of these relationships over time and include a broader cultural range to enhance the generalizability of the findings. The role of external factors such as family dynamics, peer interactions, and specific national policies surrounding technology use should be further investigated as potential moderators in understanding cyberbullying behavior. Future studies should consider incorporating more situational variables within the GAM framework to better capture the nuances of online aggression and its prevention in various cultural contexts.

## Figures and Tables

**Figure 1 ijerph-22-00389-f001:**
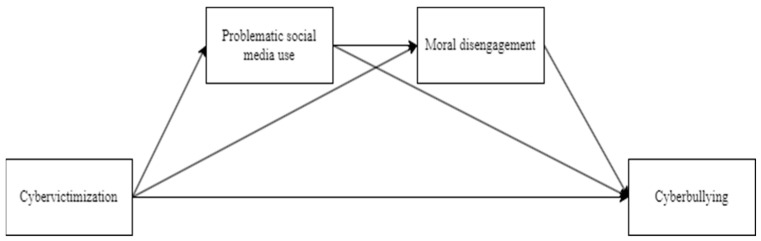
Theoretical model of serial mediation effects, linking cybervictimization and cyberbullying through the serial mediating actions of PSMU and MD.

**Figure 2 ijerph-22-00389-f002:**
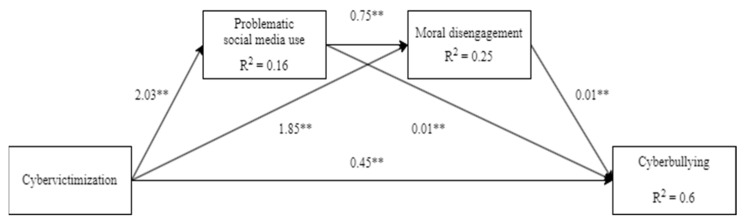
Non-standardized estimate results from the serial mediating research model. ** *p* > 0.001.

**Table 1 ijerph-22-00389-t001:** Descriptive statistics in the Italian (*n* = 489) and Spanish (*n* = 406) samples.

	Italy		Spain	
	*N*	*%*	*N*	*%*
**Gender**				
Males	236	48.3	197	48.5
Females	246	50.3	209	51.5
Prefer not to respond	6	1.2		
Non-binary	1	0.2		
**Age**				
9	-	-	44	10.8
10	14	2.9	165	40.6
11	131	26.8	182	44.8
12	236	48.3	13	3.2
13	105	21.5	2	0.5
14	3	0.6		
	** *M (SD)* **		** *M (SD)* **	
Problematic social media use	13.64 (4.84)		11.56 (4.77)	
Moral disengagement	53.3 (13.29)		44.40 (11.84)	
Cybervictimization	14.42 (4.97)		12.38 (3.37)	
Cyberbullying	13.21 (4.29)		11.53 (1.62)	
Age	11.90 (0.783)		10.42 (0.745)	

**Table 2 ijerph-22-00389-t002:** T-test results for independent samples and mean differences controlled for gender.

Variables	Females (M ± SD)	Males (M ± SD)	*t*-Value	df	*p*-Value	Mean Difference	Cohen’s d	Effect Size Interpretation
Problematic social media use	12.74 ± 5.21	12.66 ± 4.62	0.253	886	0.8	0.08	0.01	Small
Moral disengagement	46.97 ± 12.9	51.51 ± 13.4	−5.15	886	0.000	−4.54	−0.34	Medium
Cybervictimization	0.02 ± 0.82	0.09 ± 0.86	−1.28	886	0.2	−0.07	−0.08	Small
Cyberbullying	0.005 ± 0.7	0.13 ± 0.82	−2.47	848.27	0.014	−0.127	−0.16	Small

**Table 3 ijerph-22-00389-t003:** Bivariate correlations among the primary variables in the whole sample.

Variables	1	2	3	4	5
1. Problematic social media use	1				
2. Moral disengagement	0.373 ***	1			
3. Cybervictimization	0.344 ***	0.215 ***	1		
4. Cyberbullying	0.327 ***	0.274 ***	0.573 ***	1	
5. Age	0.175 **	0.373 ***	0.344 ***	0.327 ***	1

*** *p* < 0.001. ** *p* < 0.05.

**Table 4 ijerph-22-00389-t004:** Direct and indirect effects of standardized estimates of cybervictimization (CV, Independent variable), Problematic Social Media Use (PSMU), and Moral Disengagement (MD) as serial mediators of cyberbullying (CB; outcome).

Pathway	*Β*	*SE*	*t*	95% [CI]		*P*	F
				*LL*	*UL*		
**CV -> CB**							
Total effect	0.071	0.013		0.045	0.098	<0.001	
Direct effect	0.499	0.026	17.28	0.401	0.503	<0.001	85.47
**Specific indirect effects**							
CV-PSMU-MD-CB	0.013	0.003		0.006	0.021	<0.001	
CV-PSMU-CB	0.041	0.012		0.016	0.067	<0.001	44.58
CV-MD-CB	0.016	0.006		0.006	0.029	<0.001	59.36

Note: CI = Confidence Interval; LL = lower limit; UL = upper limit.

## Data Availability

The datasets presented in this article are not readily available because the data are part of an ongoing study. Requests to access the datasets should be directed to corresponding author.

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
