# Peer review of "Cyberbullying Perpetration and Socio-Behavioral Correlates in Italian and Spanish Preadolescents: A Cross-National Study and Serial Mediation Analysis"

_ijerph, 2025, doi:10.3390/ijerph22030389_

Round 1

Reviewer 1 Report

Comments and Suggestions for Authors

Peer Review Report for the manuscript “Cyberbullying Perpetration and Socio-Behavioral Correlates in Italian and Spanish Preadolescents: A Cross-National Study and Three Serial Mediation Analyses.”

The manuscript is well-written and organized. The authors present a comprehensive study in a clear and accessible manner. The topic is unique, interesting, and relevant to public health, particularly concerning a problem faced by one of our most vulnerable populations. The methodology employed in the study is scientifically sound and executed effectively.

I have a few comments that could help clarify certain aspects and enhance the overall quality of the manuscript:

1. In lines 164-165, the authors state that they adopted the term "problematic social media use" instead of "social media addiction" (SMA). However, the term “SMA” still appears throughout the paper. Please replace all instances of “SMA” with “problematic social media use.”

2. In line 68, what does HBSC stand for? Please spell out all the words.

3. In lines 236 and 238, I recommend that the authors spell the abbreviations RQ1 and RQ2 as “Research Question 1” and “Research Question 2,” respectively.

4. In line 296, the authors seem to have used a convenience sample. I suggest revising the wording to reflect this accurately.

5. In lines 295–297, I encourage the authors to provide more information regarding the data collection procedure, such as the number of students approached, those who agreed to participate, the number who completed the survey, and the overall response rate.

6. In line 297, I suggest that the authors include the proportion of Spanish students in addition to the Italian students.

7. The sentence in lines 318-319 is unclear. I recommend rephrasing it, particularly the phrase “Internet and social media platforms frequency of use and habits among the participants.”

8. Are there English versions of the scale methods (BSMAS, MDS, ECIP-Q)? If so, I suggest that the authors provide them in the Appendices.

9. The authors provided the total mean score for the MD variable. I recommend that the authors also include the mean scores for PSMU, CB, and CV.

10. In lines 365-366, the authors indicate that “Gender was dichotomous,” yet they provided four choices. I suggest that the authors clarify what happened to the cases where participants preferred not to answer or selected "other" in the data analysis.

11. In lines 382-384, the authors state that there were “no statistically significant mean differences across the variables when comparing gender groups.” However, in Table 2, the p-values for MD and CB are less than 0.05. Please explain this discrepancy.

12. In Table 3, I recommend that the authors specify “primary” or “examined” variables and explain the relevance of the variable “age” within the table.

Congratulations to the authors on their well-done work.

Author Response

Comments on revisions

Dear Reviewer,

I would like to express my deep gratitude for your thorough and insightful review of our manuscript. Your constructive feedback and thoughtful comments have been invaluable in refining and strengthening the manuscript. I deeply appreciate the time and effort you dedicated to providing such detailed and knowledgeable guidance. Your consideration has significantly contributed to the improvement of the study. All changes made to the manuscript are set out in this letter and highlighted in red in the text.

Kind regards.

Comment 1 – “Use of Problematic Social Media Use”: In lines 164-165, the authors state that they adopted the term “problematic social media use” instead of “social media addiction” (SMA). However, the term “SMA” still appears throughout the paper. Please replace all instances of “SMA” with “problematic social media use.”

Answer 1: Thank you for the comment. I agree about the need to edit all SMA instances with PSMU. I replaced the term throughout the text, tables and figures. The changes can be found at lines 321, Table 1, Table 3, Table 4.

Comment 2 – “Use of the term HBSC”: In line 68, what does HBSC stand for? Please spell out all the words.

Answer 2: Thank you for highlighting this error. I added full term in line 64-65.

Comment 3 – “Abbreviations of Research Question”: In lines 236 and 238, I recommend that the authors spell the abbreviations RQ1 and RQ2 as “Research Question 1” and “Research Question 2,” respectively.

Answer 3: Thank you for the suggestion. In lines 229 and 231, I replaced the two abbreviations with the full instances.

Comment 4 – “Use of convenience sample”: In line 296, the authors seem to have used a convenience sample. I suggest revising the wording to reflect this accurately.

Answer 4: Thank you for this comment. In line 288, I specified that we administered our survey to a convenience sample.

Comment 5 – “Information on data collection”: In lines 295–297, I encourage the authors to provide more information regarding the data collection procedure, such as the number of students approached, those who agreed to participate, the number who completed the survey, and the overall response rate.

Answer 5: Thank you for the invitation. We agree about the urgency to clarify these specific aspects of data collection procedures. Thus, in lines 290 – 291 I added: “All approached students participated in the survey in both countries, since the ones excluded did not receive consent from their parents.”.

Comment 6 – “Proportion of Spanish students”: In line 297, I suggest that the authors include the proportion of Spanish students in addition to the Italian students.

Answer 6: Thank you for the comment. In line 289, I precisely reported the percentage of Spanish students that participated in the self-report survey.

Comment 7 – “Unclear sentence on Internet and SNs use”: The sentence in lines 318-319 is unclear. I recommend rephrasing it, particularly the phrase “Internet and social media platforms frequency of use and habits among the participants.”

Answer 7: Thank you for the recommendation. I acknowledge that the phrase is difficult to read and interpret, thus I revised it in a more readable and direct format. The new form is “Sociodemographic information was collected such as gender, age and frequency of use and habits related to Internet and social media platforms.” (lines 313 – 314).

Comment 8 – “English version of scales”: Are there English versions of the scale methods (BSMAS, MDS, ECIP-Q)? If so, I suggest that the authors provide them in the Appendices.

Answer 8: Thank you for your comment. Yes, the English versions of the scales do exist; thus, we added them in a separate Supplementary materials file. I mentioned the tables in the main text as Tables S1, S2, S3 (lines 321 – 322; lines 331 – 332; line 345). I also added the references to the scales, subsequently revising all references’ number in the main text (lines 822-823; lines 831-832).

Comment 9 – “Total mean scores for variables”: The authors provided the total mean score for the MD variable. I recommend that the authors also include the mean scores for PSMU, CB, and CV.

Answer 9: Thank you for the recommendation. I agree about the importance of reporting the correct total mean score for each variable. However, we would like to clarify that the total mean scores for all variables, including PSMU, CB, and CV, are already presented in Table 1 of the manuscript. If there is any specific aspect that requires further clarification or if additional details are needed, we would be happy to address them accordingly. Please let us know how we can improve the presentation of this information to ensure clarity.

Comment 10 – “How to treat Gender variable”: In lines 365-366, the authors indicate that “Gender was dichotomous,” yet they provided four choices. I suggest that the authors clarify what happened to the cases where participants preferred not to answer or selected “other” in the data analysis.

Answer 10: Thank you for your thoughtful comment. We appreciate the opportunity to clarify this aspect of our data analysis. While we provided four response options for gender, cases where participants selected “other” or “prefer not to answer” were excluded from the analysis due to their very low number, which did not allow for meaningful statistical comparisons. We acknowledge the importance of addressing gender diversity in research and appreciate your suggestion. We will ensure that this clarification is explicitly stated in the revised manuscript. I stated it at lines 365 – 366.

Comment 11 – “Statistically significant mean differences”: In lines 382-384, the authors state that there were “no statistically significant mean differences across the variables when comparing gender groups.” However, in Table 2, the p-values for MD and CB are less than 0.05. Please explain this discrepancy.

Answer 11: Thank you for your valuable feedback. We acknowledge the discrepancy and have now clarified this point in the revised manuscript. Specifically, we added a sentence stating that males showed higher mean scores in both moral disengagement and cyberbullying (p < 0.05), suggesting they may be more involved in online aggressive interactions than females. This revision ensures alignment between our statistical findings in Table 2 and the text. We appreciate your insightful suggestion and believe this clarification strengthens the accuracy and coherence of our results. The changes are reported in lines 384 – 388.

Comment 12 – “”: In Table 3, I recommend that the authors specify “primary” or “examined” variables and explain the relevance of the variable “age” within the table.

Answer 12: Thank you for your comment. I agree about both recommendations; thus I added “primary” (line 396) and specified the importance of “age variable” within the table (lines 394 – 395).

Conclusion: Thank you. I acknowledge the importance of your comments to enhance the overall quality of this study. I have tried to review solely and exclusively to improve clarity and coherence. We have reread the article numerous times, and the work has been edited in its entirety, in some parts the language has been simplified to facilitate clarity and readability. Overall, I appreciate the feedback and strived to enhance the clarity and precision of the discussion section by explicitly linking the research findings to the theoretical framework, thereby addressing the inherent limitations of the study while providing valuable insights into the topic under investigation in the contribution.

Gianluca Mariano Colella

Rende (CS), 27/02/2025

gianluca.colella@unical.it

Department of Cultures, Education and Society, University of Calabria

Reviewer 2 Report

Comments and Suggestions for Authors

This study contributes to a deeper understanding of the mechanisms of cyberbullying perpetration among preadolescents. The manuscript is well-structured, methodologically rigorous, and framed within an established theoretical model. However, several areas need improvement, particularly in the areas of theoretical assumptions, validity of measures, and clarity of statistical data. Detailed suggestions for improving the manuscript are provided below:

  1. The study uses a cross-sectional design, but interprets the relationships in a way that suggests causality. The authors should explicitly acknowledge this limitation in the discussion and emphasize that causal relationships cannot be inferred from cross-sectional data. I suggest they consider adding a statement in the limitations section suggesting that longitudinal studies are needed to confirm the directionality of these relationships.
  2. The cross-national comparison between Italy and Spain is valuable, but the discussion lacks a thorough interpretation of cultural factors that may influence cyberbullying behaviors. Although the study mentions policy differences (e.g., the smartphone ban in Spain), it does not adequately explore how cultural norms, educational practices, or family dynamics may moderate the relationships studied. I recommend adding a subsection in the discussion that elaborates on these cultural differences, supported by additional references.
  3. The manuscript states that the cyberbullying scale (ECIP-Q) showed low reliability (α = .616 in the Spanish sample), raising questions about its validity. The authors should discuss the reason for this low reliability and how it may affect the results. They should also consider conducting an item analysis to determine if specific items are problematic. If removing the items improves reliability, the revised measure should be reported.
  4. The logic of placing PSMU before MD in the serial mediation model is not entirely justified. Although the authors refer to literature that supports this order, the relationship between problematic social media use and moral disengagement is complex and may be bidirectional. Consider revising the introduction and hypotheses to clarify why this specific model was tested versus alternative pathways (e.g., MD → PSMU → CB).
  5. The research questions (lines 235-239) and hypotheses (lines 260-264) could be more clearly stated. Consider explicitly stating the expected direction of the relationships and the role of each mediator before presenting the statistical model.
  6. The introduction contains several redundant statements about the prevalence of cyberbullying. Consider condensing this section to avoid repetition.
  7. The literature review could be expanded to include recent findings on the role of parental mediation and family dynamics in adolescents' digital behaviors. Specifically, I recommend that the authors cite and discuss these recent articles on the topic to strengthen this section of the manuscript: 10.1111/jora.13034; 10.1016/j.psychres.2017.05.030; 10.1007/s12144-023-04557-6.
  8. Implications for intervention programs are discussed, but specific recommendations are lacking. Consider including examples of successful cyberbullying prevention programs and practical suggestions for educators and policymakers based on the study findings
Comments on the Quality of English Language

The manuscript is well written, but some sentences are too complex. Consider simplifying the sentences to make them more readable.

Author Response

Comments on revisions

Dear Reviewer,

I would like to express my gratitude for your thorough and insightful review of our paper. Your feedback and thoughtful comments have been invaluable in refining and strengthening the manuscript. I deeply appreciate the time and effort you dedicated to providing such detailed and knowledgeable guidance. Your consideration has significantly contributed to the improvement of the study, and I am sincerely grateful for your contributions. All changes made to the manuscript are set out in this letter and highlighted in red in the text.

Kind regards.

Comment 1 – “Use of cross-sectional design and causality”: 1. The study uses a cross-sectional design but interprets the relationships in a way that suggests causality. The authors should explicitly acknowledge this limitation in the discussion and emphasize that causal relationships cannot be inferred from cross-sectional data. I suggest they consider adding a statement in the limitations section suggesting that longitudinal studies are needed to confirm the directionality of these relationships.

Answer 1: Thank you for your recommendation. We agree about the need to acknowledge the limitation on the use of a cross-sectional design and the impossibility to infer causality. Thus, in lines 539 - 552 we already addressed such criticism, recognizing the need for longitudinal studies is important. We added a statement at lines 556 - 557 to encourage the adoption of longitudinal studies:

“While mediation analysis is a useful statistical technique, it does not allow robust testing of causality when bidirectional and simultaneous influences likely exist among the variables. Specifically, in the relationship between CV and CB, there is strong theoretical justification for conceptualizing cybervictimization as a situational trigger that elicits impulsive aggressive behaviors, a mechanism often observed in aversive situations where hostile attitudes are activated. This view suggests a more immediate, reactive link between CV and CB. The role of PSMU may create tension within the hypothesized temporal sequence, as impulsive aggressive behaviors theoretically contradict the sustained behavioral engagement associated with PSMU. However, the model and relationships tested, and the results obtained confirm similar results obtained from longitudinal studies on these variables.”

“However, future research should employ longitudinal data to better assess the directionality of these relationships.”

Comment 2 – “Cross-national differences”: The cross-national comparison between Italy and Spain is valuable, but the discussion lacks a thorough interpretation of cultural factors that may influence cyberbullying behaviors. Although the study mentions policy differences (e.g., the smartphone bans in specific Spain regions), it does not adequately explore how cultural norms, educational practices, or family dynamics may moderate the relationships studied. I recommend adding a subsection to the discussion that elaborates on these cultural differences, supported by additional references.

Answer 2: Thank you for your comment. We completely agree that cultural factors, beyond policy differences, may play a crucial role in shaping cyberbullying behaviors across Italy and Spain. In response to your suggestion, we have edited the discussion section, addressing how cultural norms, educational practices, and family dynamics in Italy and Spain may moderate the studied relationships. Additionally, we have integrated further references to support this discussion (lines 449 - 460):

“Cultural, educational, and policy-driven factors may influence the differences between Italian and Spanish participants in cyberbullying-related variables. One key difference lies in educational policies regarding technology use. Such policy variations between Italian and Spanish schools may contribute to differences in PSMU and online aggression patterns [102]. Beyond policy, cultural attitudes toward digital supervision and parenting styles may also play a role. Research suggests that Mediterranean parenting styles, while traditionally protective, vary across countries in terms of digital monitoring. Spanish parents tend to emphasize structured family routines and digital literacy, whereas Italian parents may lean toward greater autonomy in online interactions [103]. Future research should further explore these cultural moderators, considering how national policies, educational strategies, and familial norms interact to in-fluence cyberbullying dynamics in different contexts, as well as motivations driving online behaviors [32 – 34].”

Comment 3 – “ECIP-Q low reliability”: The manuscript states that the cyberbullying scale (ECIP-Q) showed low reliability (α = .616 in the Spanish sample), raising questions about its validity. The authors should discuss the reason for this low reliability and how it may affect the results. They should also consider conducting an item analysis to determine if specific items are problematic. If removing the items improves reliability, the revised measure should be reported.

Answer 3: Thank you for your comment. I agree with the low reliability of the ECIP-Q, and we tried to discuss the possible causes of this result in the limitations (lines 569 - 572). However, we decided not to conduct an item analysis to remove items, since we identified cultural causes for these results (less frequent use of smartphones in Spanish preadolescents).

“Moreover, the lower reliability of the ECIP-Q among Spanish preadolescents on the cyberbullying perpetration items may be influenced by these cultural tendencies, affecting online aggressive behavior and general normative habits regarding smartphone and Internet use.”

Comment 4 – “serial mediation PSMU à MD”: The logic of placing PSMU before MD in the serial mediation model is not entirely justified. Although the authors refer to literature that supports this order, the relationship between problematic social media use and moral disengagement is complex and may be bidirectional. Consider revising the introduction and hypotheses to clarify why this specific model was tested versus alternative pathways (e.g., MD → PSMU → CB).

Answer 4: Thank you for your thoughtful comment regarding the order of variables in our serial mediation model. We acknowledge the complexity of the relationship between PSMU and MD and the potential for bidirectionality. We improved the main text as follows (lines 221 – 227):

“The relationship between PSMU and MD is based on the premise that a generalized problematic engagement with social media might precede and facilitate disengaged moral reasoning, rather than the other way around. Specifically, PSMU may act as a precursor to MD, as unregulated social media use can foster detachment from consequences, normalization of harmful behaviors, and reduced self-regulation in online social interactions. Also, the MD-PSMU causal relationship has been less explored in psychological research and was not an aim of the current study.”

Comment 5 – “Research questions and hypotheses”: 5. The research questions (lines 235-239) and hypotheses (lines 260-264) could be more clearly stated. Consider explicitly stating the expected direction of the relationships and the role of each mediator before presenting the statistical model.

Answer 5: Thank you for your comment. We explicitly stated the direction of the relationships among the variables and the role of each mediator. In lines 229 and 231 we revised the research questions, adding the expected direction of the relationship, and in lines 254 and 256 we revised the hypotheses of the study. We also edited the hypotheses when mentioned in the Discussion.

“Research Question 1: Is cybervictimization directly and positively associated to cyberbullying perpetration among Italian and Spanish preadolescents?

Research Question 2: Is this relationship mediated by the serial indirect and positive effect of PSMU and MD in both samples?”

“The hypotheses under investigation in this cross-national study are the following: (i) cybervictimization and cyberbullying perpetration present a positive direct association in a cross-national sample of Italian and Spanish preadolescents; (ii) PSMU and MD positively and serially mediate the relationship between cybervictimization and cyberbullying perpetration in Italian and Spanish preadolescents.”

Comment 6 – “Condensing the introduction”: The introduction contains several redundant statements about the prevalence of cyberbullying. Consider condensing this section to avoid repetition.

Answer 6: Thank you for this recommendation. I agree about the need to sum up redundant statements. Thus, I erased some parts and condensed the section. I highlighted in red the changes, presenting a shorter introduction.

Comment 7 – “Enhancing references”: The literature review could be expanded to include recent findings on the role of parental mediation and family dynamics in adolescents' digital behaviors. Specifically, I recommend that the authors cite and discuss these recent articles on the topic to strengthen this section of the manuscript: 10.1111/jora.13034; 10.1016/j.psychres.2017.05.030; 10.1007/s12144-023-04557-6.

Answer 7: Thank you for this suggestion. I agree about the need to shed light on the role of parental mediation and family dynamics in preadolescents’ online experiences. Thus, I added the three references (lines 99 - 101; lines 675 - 681) to enhance quality of the introduction and discussed them in the specific section when addressing the role of parenting mediation in cyberbullying behaviors across Italy and Spain (line 460).

“Among the possible risks faced by preadolescents, problematic social media, videogames and Internet use have progressively emerged as potentially impairing pressing public health concerns across numerous Western countries, due to their im-pact on preadolescents’ social development processes [26 – 31]. Particularly, in association with parental monitoring attitudes, recent studies illustrated the interplay be-tween escapism motivation to reduce negative emotions, parental mediation styles, and interpersonal skills [32 - 34].”

Comment 8 – “Practical implications”: Implications for intervention programs are discussed, but specific recommendations are lacking. Consider including examples of successful cyberbullying prevention programs and practical suggestions for educators and policymakers based on the study findings.

Answer 8: Thank you for this recommendation. In the conclusion, I added statements on EBPs and research-interventions to counteract cyberbullying, adding specific references (lines 508 – 512; lines 863 - 869). Thus, we hope to satisfy the need to provide practical suggestions about cyberbullying prevention programs.

“Based on our findings, and on previous literature, educators and policymakers might consider adopting evidence-based research-interventions to practically address cyberbullying perpetration and suffering, as well as negative online habits in preadolescents and adolescents, through socio-emotional and moral learning, peer culture education, or digital literacy [106 - 108].”

Conclusion: Thank you. I acknowledge the importance of your comments to enhance the overall quality of this quantitative study both theoretically and practically. I have tried to comment on the introduction and discussion solely and exclusively to improve clarity and coherence regarding our choices but also including your reviews. We have reread the article numerous times, and the work has been edited in its entirety, in some parts the language has been simplified to facilitate clarity and readability. Overall, I appreciate the feedback and strived to enhance the clarity and precision of the discussion section by explicitly linking the research findings to the theoretical framework, thereby addressing the inherent limitations of the study while providing valuable insights into the topic under investigation in the contribution.

Gianluca Mariano Colella

Rende (CS), 27/02/2025

gianluca.colella@unical.it

Department of Cultures, Education and Society, University of Calabria

Round 2

Reviewer 2 Report

Comments and Suggestions for Authors

I have no further revisions to recommend.